# Modulation of Bovine Endometrial Cell Receptors and Signaling Pathways as a Nanotherapeutic Exploration against Dairy Cow Postpartum Endometritis

**DOI:** 10.3390/ani11061516

**Published:** 2021-05-23

**Authors:** Ayodele Olaolu Oladejo, Yajuan Li, Xiaohu Wu, Bereket Habte Imam, Jie Yang, Xiaoyu Ma, Zuoting Yan, Shengyi Wang

**Affiliations:** 1Key Laboratory of Veterinary Pharmaceutical Development of Ministry of Agriculture, Lanzhou Institute of Husbandry and Pharmaceutical Sciences of Chinese Academy of Agricultural Science, Lanzhou 730050, China; aygenesis2succeed@yahoo.com (A.O.O.); 18719826933@163.com (Y.L.); wuxiaohu01@caas.cn (X.W.); Bekihi14@gmail.com (B.H.I.); 15535384771@163.com (J.Y.); 18903865563@163.com (X.M.); 2Department of Animal Health Technology, Oyo State College of Agriculture and Technology, Igboora 201103, Nigeria

**Keywords:** endometritis, cell receptors, signaling pathways, PAMP, biomolecules, cytokines, nanotherapy, dairy cow, nanosystem

## Abstract

**Simple Summary:**

The provision of updated information on the molecular pathogenesis of bovine endometritis with host-pathogen interactions and the possibility of exploring the cellular sensors mechanism in a nanotechnology-based drug delivery system against persistent endometritis were reported in this review. The mechanism of Gram-negative bacteria and their ligands has been vividly explored, with the paucity of research detail on Gram-positive bacteria in bovine endometritis. The function of cell receptors, biomolecules proteins, and sensors were reportedly essential in transferring signals into cell signaling pathways to induce immuno-inflammatory responses by elevating pro-inflammatory cytokines. Therefore, understanding endometrial cellular components and signaling mechanisms across pathogenesis are essential for nanotherapeutic exploration against bovine endometritis. The nanotherapeutic discovery that could inhibit infectious signals at the various cell receptors and signal transduction levels, interfering with transcription factors activation and pro-inflammatory cytokines and gene expression, significantly halts endometritis.

**Abstract:**

In order to control and prevent bovine endometritis, there is a need to understand the molecular pathogenesis of the infectious disease. Bovine endometrium is usually invaded by a massive mobilization of microorganisms, especially bacteria, during postpartum dairy cows. Several reports have implicated the Gram-negative bacteria in the pathogenesis of bovine endometritis, with information dearth on the potentials of Gram-positive bacteria and their endotoxins. The invasive bacteria and their ligands pass through cellular receptors such as TLRs, NLRs, and biomolecular proteins of cells activate the specific receptors, which spontaneously stimulates cellular signaling pathways like MAPK, NF-kB and sequentially triggers upregulation of pro-inflammatory cytokines. The cascade of inflammatory induction involves a dual signaling pathway; the transcription factor NF-κB is released from its inhibitory molecule and can bind to various inflammatory genes promoter. The MAPK pathways are concomitantly activated, leading to specific phosphorylation of the NF-κB. The provision of detailed information on the molecular pathomechanism of bovine endometritis with the interaction between host endometrial cells and invasive bacteria in this review would widen the gap of exploring the potential of receptors and signal transduction pathways in nanotechnology-based drug delivery system. The nanotherapeutic discovery of endometrial cell receptors, signal transduction pathway, and cell biomolecules inhibitors could be developed for strategic inhibition of infectious signals at the various cell receptors and signal transduction levels, interfering on transcription factors activation and pro-inflammatory cytokines and genes expression, which may significantly protect endometrium against postpartum microbial invasion.

## 1. Introduction

Researchers have reported several reports on reducing dairy cattle reproductive efficiency because of postpartum development of uterine disease or infection in dairy cows [1,2,3,4]. Several diagnostic, treatment and prevention such as purulent vulval discharges, return to oestrus after breeding, presence of microorganism from microbiological examination, infiltration of polymorphonuclear cells, hormonal variation, antibiotic and anti-inflammatory treatment regimen, cleanliness of cattle environment, avoidance of coitus with infected animals, fomite sterilization and vaccinationhave been adopted to eliminate this postpartum uterine infection, which includes rectal palpation of the genital tract to detect uterine abnormalities [3,5,6,7,8,9,10]. Postpartum endometritis of dairy cows usually occurs, causing economic losses due to high infectivity percentages, which has been a significant concern to dairy scientists, particularly at the onset of this millennium [3,11]. Postpartum uterine disease reflects disturbance on the normal physiological characteristics during postpartum in the dairy cow uterus, leading to impaired uterine involution. Monitoring the uterine involution, endometrial restoration, resumption of ovarian cyclicity, and pathogenic bacteria control are the major postpartum activities to ensure continuous reproductive ability [6,12]. Postpartum massive pathogens invasion of endometrium leads to the incapability of the uterine immune system to combat pathogens. The inflammatory response to postpartum bacterial infection during subclinical endometritis is correlated with the secretion of pro-inflammatory mediators that cause cell proliferation by either reacting on the oocyte, forming the embryo, endometrium, or impairing the hypothalamic-pituitary-gonadal axis [4]. The anatomical position and vascularization of the endometrium are physiological arrangements that make it prone to override by bacteria invasion. Endometritis occurs as a result of its inadequacy to disrupt pathogen growth and multiplication in the uterine epithelium. *Escherichia coli* are the most abundant endometrial pathogenic bacteria interacting with other bacteria isolated from cows with uterine disease [13,14].

In addition to bacteria, the in vitro infection of bovine epithelial cells with bovine herpesvirus IV [BoHV-4] has been reported suggesting the endometrium could be susceptible to viral infection [12]. However, the molecular mechanism implicated in the pathogenesis of endometritis has not been deeply investigated. The innate immune system recognizes pathogenic bacteria that invade the endometrial tissue by host cell PRRs to bacteria PAMPs [14]. PAMPs like LPS, a major component of cell walls unique to most gram-negative bacteria, stimulate immune system cells and induce a strong inflammatory response [15]. The inflammatory intracellular signaling pathway cascades induced by the interactions between PAMPs and PRRs contribute to the transcription and release of crucial pro-inflammatory mediators, including IL-β, TNFα, and chemokines responsible for complement activation and acute phase protein response that results in successful immune and inflammatory responses to eradicate the pathogens [16,17]. However, chronic uterine inflammation attributable to insufficient uterine pathogen eradication, excessive inflammatory signals, and anti-inflammatory pathway deficiencies is likely to affect fertilization and pregnancy [18], ultimately leading to subclinical endometritis. It was reported that early postpartum endometrial inflammation is a normal physiological process aimed at repairing the damaged endometrial uterine lining, though consequently bacteria [7] and their ligands invade the lumen, resulting in disruption of the typical anatomical architecture of the uterine mucosa, leading to endometritis. The quest to provide an in-depth review highlighting the understanding of the mechanism of dairy cow endometritis pathophysiology through synergy existing between bacterial PAMPs, activation of PRRs of the endometrial cells, cellular signaling pathways activation and phosphorylation co-exist, leading to production and regulation of inflammatory cytokines, which are mediators to the onset of bovine postpartum endometritis. Therefore, there are critical needs to evaluate the pathogenesis cascade of endometritis and endometrial cell sensors and biomolecules. We reviewed the current knowledge and research article about cell molecular characterization that may be affecting the endometrium being prone to microbial invasion during postpartum and a possible pointer tool for nanodrug development or immunotherapeutic index in curbing the persistence of dairy cow endometritis.

## 2. Function of Bacterial Pampsin Bovine Endometritis

Bacterial identification in mammalian tissues relies on the perception of their PAMP on innate immune system cells. PAMP recognition of PRRs triggers a cascade of events that cause host protection mechanisms to deter and resist the initial infection and prompt immune response. Endogenous molecules and PAMPs may directly tie to the pathogenesis of endometrial diseases [19,20,21,22]. The Gram-positive bacteria cell wall comprises a thick layer of peptidoglycan mixed with teichoic acid and extracellular proteins. Some PAMPs produced by Gram-positive bacteria are Lipoteichoic Acid [LTA], lipoproteins, peptidoglycan, glycosyl phosphatidyl, hemagglutinin protein, and phospholipomannan [20,21,22]. The Gram-negative bacteria cell wall comprises a small, elastic protein peptidoglycan layer coated in a PAMP additional layer [23]. The Gram-negative bacteria produce PAMP capable of inducing inflammatory reaction activating specific PRR and phosphorylation of relevant cellular signaling pathways [24,25], stimulating the immune response. PAMPs are subject to evolutionary change due to selective pressure from host defense mechanisms. This discovery may eventually expand the range of ligands to a given PRR and the cellular signals it can elicit [25]. 

Most unique PAMPs are identified by various PRRs to synergistically cause inflammatory responses and guide the adaptive immune responses [22]. Some ligands expressed their function either on the cell wall or cytoplasm and other subcellular organelles of the host cell. Evaluating the pathogenesis of bacteria causing dairy cow endometritis needs to review the role of the entire Gram-negative bacteria endotoxins and virulence factors in penetrating the cell and activating cellular function. *Escherichia coli* produces cell-wall bacterial components such as LPS [26], cholesterol-dependent cytotoxin and pyolysin, *Truperella pyogenes*, and *Fusobacterium necrophorum* produce growth factor F and Leukotoxin; whereas *Provetella* species develops a phagocytose-inhibiting agent [27]. The most abundant Gram-negative PAMP that can induce inflammatory and immune responses of invaded endometrial cells is LPS, a major component of Gram-negative bacteria external membrane [28]. LPS from *E. coli* is the principal Gram-negative bacteria PAMP that has been evaluated in endometritis pathomechanisms [11,26]. However, the molecular mechanisms of other Gram-negative bacteria to induce endometritis remain untapped. 

Hexa-acetylated lipid A types present in *E.coli* and *Salmonella enterica serovar Typhimurium* are effective cell receptor activators [27]. As shown in experimental models of *Salmonella canoni, Legionella pneumophila*, and *E.coli* infections [29,30], the flagellin component of specific gram-negative cell walls is potent PAMPS. Still, their role in bovine endometritis pathogenesis remains unknown. *E. coli* flagella [H serogroup], Shiga-toxin [a heat-stable and labile toxin], and lipopolysaccharide [O serogroup LPS] [31,32], fimbriae, endotoxigenic of other Gram-negative bacteria cell wall proteins and adhesins are putative ligands for various PRRs. 

At the time of this review’s compilation, studies on the molecular and cellular evaluation of dairy cow endometrium inflammatory response after calving due to invasion of Gram-positive bacteria and their ligand endotoxins were absent. The PAMPs, bacterial load, and virulence factors of the postpartum endometrial pathogens determined whether infection of the postpartum uterus evolves toward inflammatory conditions like clinical or subclinical endometritis [25,32]. Certain Gram-positive bacteria, including *Corynebacterium, Nocardia, Bacillus, Listeria, Staphylococcus, Clostridium*, *Enterococcus, Streptococcus,* and *Mycobacterium species* are some of the pathogens implicated in postpartum endometritis pathogenesis but not specifically characterized ding uterine infection [21,33]. They produced PAMPs such as lipopeptides, peptidoglycan, glycolipids, LTA [34], though it remains elucidated whether the upregulation of those specific PRRs can induce immuno-inflammatory responses.

Triacylated lipoprotein from a Gram-positive bacterium has been shown to stimulate an inflammatory response by activating a specific PRR with subsequent phosphorylation and activation of the intracellular signaling pathway [35,36]. The apparent dearth of studies on the impact of Gram-positive bacteria and their ligands on the pathogenesis of postpartum endometritis in dairy cows, as it was reported [37] suggesting that these pathogens and their PAMPs may have a long-term retarding effect on the current method used in the evaluation of postpartum uterine immune suppression related to pro-inflammatory cytokines and chemokines activation induced by uterine exposure to the influx of bacteria and their ligands. Hence, the concern for prompt research actions in postpartum dairy cows. The mechanism through which the bacteria and PAMPs penetrate the various cell receptors to induce cellular and molecular changes needs to be in the pathogenesis of bovine endometritis as it is not thoroughly analyzed up today.

## 3. Endometrial Cell Receptors and Signalling Biomolecules

The bovine endometrium epithelial and stromal cells have been reported to have multiple PRRs, which normally identify PAMPs that contribute to the stimulation of cell signaling pathways to enhanced production of numerous inflammatory mediators correlated with endometrial inflammatory disease pathogenes is [37,38]. The review of patterns by which different endometrial cell receptors recognize various signals from microorganisms and their ligands in responsive actions leading to postpartum dairy cow endometritis. Several pathogen receptors have been detected in endometrial cells involved in immuno-inflammatory reaction complexes [39]. Each of the endometrial PRRs had an affinity for different bacteria PAMPs and cellular locations [40].

TLRs are cell membrane-bound proteins with an additional transmembrane domain implicated in ligand recognition, whether on the extracellular surface, within endosomes, and cytoplasmic domain engaged in signal transduction [38,41]. NLRs and RLRs are cytosolic helicase receptors able to detect PAMPs’ cellular invasion [21]. TLRs and NLRs are major PRRs identified in endometrial cells, synergize and complement each other intracellularly [42,43], but the mechanism through which their synergism results in the activation and phosphorylation of intracellular signaling pathways remain unknown. Bovine endometrial revealed in healthy non-gravid cows, the uterus ipsilateral and contralateral to the corpus luteum horn have expressed TLR1 to TLR10 [43,44]; TLR2, TLR3, TLR4, and TLR6 are expressed ante- and postpartum in caruncular and inter-caruncular areas of the endometrium, and whereas TLR9 has a greater expression in caruncular areas [44]. Also, endometrial epithelial cells in culture expressed TLR1, TLR7and TLR9 [45], and stromal cells expressed TLR1, TLR4, TLR6, TLR7, TLR9, and TLR10 [45,46]. TLRs are divided into two subgroups regarding their cellular localization and PAMP detection, in which TLR1, TLR2, TLR4, TLR5, TLR6, and TLR10 are expressed exclusively on the cell surface and stimulated by lipids, lipoproteins, and proteins of pathogen membrane components [46] and TLR3, TLR7, Ref. [47] and TLR9 are expressed on intra-cellular organelle membranes such as endosomes (Figure 1) [26,47,48,49]. The negative feedback mechanism of TLR regulation results in aberrant signaling by improper identification of self-proteins as foreign antigens, thus, resulting in excessive inflammation, systemic infectious diseases, and autoimmune diseases [43,50,51,52].

TLR signaling pathways contain signaling biomolecules and adaptor proteins, including MyD88, Mal/TIRAP, TRIF, TRAM, and SARM proteins, cell-specific and species-specific [39,49]. Deep cellular and molecular knowledge of TLRs and their biomolecule proteins function may lead to their role as therapeutic targets for medical application. TLRs are studied predominantly in uterine infection and endometritis, but their physiological role in other female reproductive events needs further investigation [50]. TLR2 detects invasion of bacterial and mycobacterial triacylated lipopeptides, mycoplasma diacylated lipopeptides, peptidoglycan [PGN], and LTA from gram-positive bacteria [34]. The bovine endometrial epithelial and stromal cells culture stimulated by diacylated and triacylated bacterial lipopeptides activates TLR2 leading to increased production of inflammatory cytokines infiltrating leukocytes. Still, the signaling pathway through which the expression occurred remains unsolved [37,51]. The TLR1–TLR2, TLR2–TLR6 heterodimeric interactions within adaptor protein, and hydrophobic interactions are responsible for increased immuno-inflammatory response [52]. TLR1–TLR10 formed dimers from the same subfamily [53], while TLR4–TLR6 heterodimer is formed in reaction to endogenous ligands established to foster inflammatory responses [51]. The intracellular immuno-signaling within subcellular organelles regulates the extend of TLR 2, 4, or 6 in immune cells, evoke innate and cellular immune responses and promote lysosomal degradation [53,54]. TLR4 signaling complex is a complete cell receptor that enhanced signal transfer within the cell and its micromolecules to elicit an inflammatory response via the transcription of cytokine genes [54,55] and the most examined receptors in bovine endometrium [40]. The TLR4 signaling pathway contributes to the development of different pro-inflammatory cytokines and chemokines linked to inflammatory responses. Immunological assays of TLRs will be vital for improving our understanding of early inflammatory events regulating immunological response due to endometritis [55]. The NLR family is the second group of endometrial cytoplasmic receptors that recognize microbial products and pathogen-associated molecular patterns. These proteins shared a common domain in organization with an NH2- terminal protein-protein interaction domain, NOD domain, and COOH-terminal leucine-rich repeats (LRR). NLRs are an active category of intracellular receptors, a key component of the hosts’ innate immune systems, which includes NOD1, NOD2, NALP, LRR, and PYD protein-containing domains 3 [56] (Figure 2). 

This review analyzed NLRs expression in dairy cow endometrial cells both in the normal and diseased state. NLRs protein expression in endometrial cells exposed to PAMPs, receptor–receptor connectivity, interaction with various cellular signaling enhanced the pathophysiology of the endometrial inflammatory response, and subsequent prolonged effect leads to subclinical endometritis. The NLR family recognizes PAMPs and biomolecules in the cytosol. NOD1 (also named as NLRC1) and NOD2 [NLRC2] are most characterized by NLR family members [56,57], which sense bacterial molecules derived from the synthesis and degradation of peptidoglycan [PGN], but its reactivity has not been explored using other bacteria ligands, especially pathogenic bacteria causing postpartum bovine endometritis. NOD1 and NOD2 identify intracellular PAMPs, and NLRP3 reacts to multiple stimuli to form a multi-protein complex.NOD1 is active in the intracellular recognition of different pathogenic bacteria, including entero-invasive *E coli.* The activation of NOD1 and NOD2 results in a signaling cascade, triggering an inflammatory response resulting in increased cytokine production [58,59]. The mechanisms by which it also delivers PGN into the cytosol to gain access to NOD1 and NOD2 are unclear. NODs and NLRP3 are essential to cell cytoplasmic-to-cytoplasmic interactions receptors, which recognize endogenous molecules and microbial molecules [60]. The molecular analysis of NLRs is defined by a centrally located NOD that induces oligomerization, a C-terminal LRR that mediates ligand with TLRs, and an N-terminal CARD responsible for the initiation of signaling [56]. If NOD1 and NOD2 are specific PAMP receptors or sense shifts in host factors resulting from the existence of microbial molecules in the cytosol is currently unclear [61]. NOD1 and NOD2 stimulation results primarily in the activation of pro-inflammatory gene expression. Other NLR proteins are involved in the activation of caspases [62]. NOD1 recognizes gD-glutamyl-meso-diaminopimelic acid [iE-DAP] present in all Gram-negative PGN structures andvarious Gram-positive bacteria, such as *Bacillus subtilis* and *L. cytogenes*. At the same time, NOD2 detects muramyl dipeptide [MDP], the largest component of the PGN motif often found in all Gram-negative and Gram-positive bacteria. 

Meanwhile, the evaluation of the pathophysiological function of PGN in subclinical endometritis has not been elucidated. A rare and essential antibacterial role in intestinal cells, such as control of antimicrobial peptides, was documented in NOD1 and NOD2 receptors [62,63]. Schroder & Tschopp 2010 [64] stated that activation of NLRs contributes to developing a multi-protein inflammasome complex, characteristically attached to caspase-1, which activates pro-inflammatory cytokine production.The molecular role of caspase-1 in the progressing onset of endometrial inflammatory response remains unevaluated. NLRs and their inflammasome are innate immune receptors that gained increased interest over the past years and are considered main intracellular pathogen sensors, and danger signals played an important role in infection and immunity [65]. It usually stimulates bovine endometrial inflammasome through molecular activities of caspase-4 dependent cascade called non-canonical inflammasome signaling pathway [65,66]. NLRP1 [NALP1] requires a protein complex dimeric reaction with NOD2 to mediate caspase-1 activation in response to MDP recognition [62,67]. NLRP2 was characterized as a maternal-specific gene in oocytes and granulosa cells, and its loss in zygotes triggered early embryonic death [68,69]. NLRP2 activation has been described in the central nervous system immune cells [70] and stimulated pro-inflammatory caspase 1, which results in pro-inflammatory cytokine and chemokine production. It has also been implicated in untoward anti-fetal responses through suppression of NF–κBsignaling pathway in linkage with the subcortical maternal complex to fertility issues were reported by Mahadevan et al. 2017 [71]. NLRP 2 activity was reported to be essential in reproductive potential and infertility [71], with no available record as at the moment of compiling this review on bovine endometritis. The NLRP3 inflammasome comprises NLRP3, ASC, and caspase-1, which are critical regulators of multiple inflammatory diseases [72].

IPAF belong to the NLR protein family and contains 22 members [73] and reacted to the exposure of endogenous ligands such as PAMPs and DAMPs to seldomly stimulates inflammatory responses in which the dysregulation in the function of NLRP3 was associated with the pathogenesis of several inflammatory diseases was reported [74,75]. Caspase1 was implicated in controlling the release of the inflammasome substrate, which is a signal produced to amplify the release of PAMPs by activating caspase-1 in neighbor cells [73,75]. The molecular mechanism and pathways through which the inflammasome complex could be activated in the endometrial remain undocumented. Several inflammasome families were identified in different body tissues and can recognize potentially harmful signals or PAMPs through their respective signaled cell receptors [57]. The NLRP3 inflammasome was reportedly formed by various PAMPs exposure of the inflicted cells with various structures. NLRP3 is majorly required for caspase-1 activation in response to bacteria ligands such as LPS, dsRNA, PGN, and LTAs when stimulated together with extracellular ATP [65].

As mentioned above, these NLRP3 functions could be explored to understand bovine endometritis pathogenesis with exposure to the PAMPs. The ionic flux, mitochondrial dysfunction, reactive oxygen species [ROS] generation, and lysosomal damage are molecular or cellular events shown to activate the NLRP3 inflammasome [76,77].

There is a need to investigate their roles in the pathogenesis of bovine endometritis. Nakahira et al. 2014 [78] reported dysfunctional mitochondria to enhance the production of mtROS to activate inflammatory activation of NLRP3 in response to LPS and ATP, which implies that NLRP3 senses cellular stress and culminate in activation of inflammasome remain to be fully elucidated, so the mitochondrial DNA [mtDNA] released into both NLRP3 and mtROS-dependent cytosols, mtDNA also interacts with both NLRP3 and AIM2, which are essential for the activation of the inflammasome during infectious phases [70,75]. Therefore, thiscould be an insight for a probable function of mtDNA of the endometrial cells’ immuno-inflammatory responses upon cell exposure to inflicting pathogens and their ligands, which at present remain uninvestigated. The urgent quest of molecular research into NLRP3 inflammatory activation of mitochondrial dysfunction, mtROS, and mtDNA could be pharmacological helpful [79]. The mechanism through which mitochondrion activates NLRP3 in the cellular pathway of bovine endometrium during prolonged inflammatory reaction needs further verification; probably it could give an insight to immunotherapeutic against incessant dairy cow endometritis.NLRP5 was expressed in mice autoantigen-specific oocyte linked with autoimmune premature ovarian failure in an infectious reproductive tract [80]. The intracellular NLRP5 in oocyte mitochondria and nucleoli near the nuclear pores show cytoplasmic and nuclear functions [79]. Decrease or absence of NLRP5 in women of advanced reproductive age result in impaired fertility and mediated mitochondrion function in mouse’s oocytes and embryo and localization of this protein in the reproductive cells, potentiating increase ROS production, depressed the cell morphology and physiology [81], but the molecular role of this inflammasome still call for concern in the molecular pathophysiology of bovine endometritis with addition to the NLRP6 inflammasomes. Likewise, the NLRP6 inflammasome was reported to promote the production and repair of intestinal epithelial cells of IL-18 as a response to inflammatory-induced intestinal wounds [82,83]. The NLRP6 is expressed in macrophages to reduce intestinal inflammation caused by bacteria to induce inflammatory cytokines expression [84,85]. Transcripts of NLRP7 were found in various human tissues, including endometrium, placenta, hematopoietic cells, all oocyte levels, and preimplantation embryos [86,87]. NLRP7 acts in chromatin reprogramming and DNA methylation across germline and premature quiescent before inflammatory response, leading to high cytokine production levels due to chorionic gonadotropin hormone related to abortion [88]. A network of biomolecules and sensors controls activation of NLRs inflammasomes in pathological conditions within NF–κBand MAPK signaling during oxidative stress and inflammatory stimuli [72]. The identified NLRs pathomechanism functions are poorly understood in innate immunity and the transcriptional factors regulation. The precise molecular mechanisms of some NLRs in bovine endometrial cells are unknown. Several reports have it that reception of PAMPs signals by cell receptors may be transported onto the cell signaling pathway depending on the immuno-inflammatory potency of the cells and their environment.

## 4. Evaluation of MAPKSignaling Transduction Pathways in Endometritis Pathogenesis

Mitogen-activated protein kinase (MAPK) pathways are implicated in several cellular processes, including proliferation, differentiation, apoptosis, cell survival, cell motility, metabolism, stress response, and inflammation. The conscious reflection of existing knowledge about cellular mechanisms adopted by pathogens and their PAMPs to target the host’s cellular MAPK signaling pathways; specialized cell receptors and highjack the immune response in dairy cow endometrium, in the manner of promoting enabling parasite maintenance in the host to induce clinical/ subclinical endometritis [52,89,90,91]. The MAPKs are signaling cascades that include various extracellular stimuli when the inflammatory response is initiated, including the development of pro-inflammatory cytokines and their substrates [90,91]. Seven MAPK families, ERK1/2, ERK3/4, ERK5, ERK7/8, NLK, C-JUN, and p38 groups, have been documented in mammalian cells [92,93]. There are two subgroups; the classical MAPKs of ERK1/2, p38, JNK, ERK5, and the atypical MAPKs of ERK3, ERK4, ERK7, and NLK [94], and are independent of or interacting with each other. Three well-known MAPK pathways are the ERK1/2, JNK1/2/3, p38 MAPK, and their isoforms, which are grouped based on their activation motif, structure, and function [95,96]. ERK1/2 is induced in response to growth factors, hormones, and pro-inflammatory stimuli, whereas JNK1/2/3 and p38 MAPKs are activated by cell environmental stresses and resultant inflammatory processes [96,97,98]. The atypical ERK remains elusive in cellular activities and disease reactions. ERK3 and ERK4 are predominant in the gastrointestinal tract and colon, respectively [99]. The elevated phosphorylation of MAPK proteins (ERK1/2, p38, and JNK) in LPS-induced endometritis with upregulated expression of pro-inflammatory cytokines and chemokines was reported [92]. MAPK phosphatases such as MKP-M targets JNK primarily, while others have a broader range, such as MKP-1 acting on most MAPKs [100]. ERK1/2 is the most studied in physiological tissues and cells, inhibiting the MAP2 K, MEK, phosphorylating the threonine and tyrosine residues in the TEY domain ERK1/2 [101]. The MAPKs share a similar organizational structure, but extracellular stress factors mainly regulate p38MAPK and JNK. ERK is preferably a target for mitogenic stimuli. The JNKs were characterized first because of their activation in response to various extracellular pressures and phosphorylate N—terminal transactivation domain transcription factors and cell regulatory in response to protein synthesis suppression [98,99,100]. The JNK1 and 2 are ubiquitous; JNK3 is confined to the brain. JNKs regulation is complex and influenced by many MKKs, with several MAPKKs activated, which are the same as p38, the TAK1, MEKK1/4, and ASK1. They can, however, activate JNK to phosphorylate MEK4 or 7 through the MAPKKs, which are MAPKKs specific to JNK [92,97,99] p38MAPK comprises four protein isoforms termed α, β, γ, and δ [98,102]. The p38δ is expressed in lung, kidney, testis, pancreas, other reproductive organs, and small intestine, p38γ is expressed in skeletal muscle and, p38α, and p38β are ubiquitously expressed [103,104]. Stimulating cells with LPS evidenced the expression of alpha-isoform of p38MAPK involved in the synthesis of pro-inflammatory cytokines. The action of cellular receptor and sequence leading to transduction of cellular signaling remains unverified. The p38β shared 50%, 63%, and 57% structural homology with p38α, p38γ, and p38δMAPK respectively.

All p38MAPK isoforms reportedly shared activation cascade with not exclusively upstream kinases targeting common molecular substrate to validate the transcripts. There are also differences among the isoforms concerning their mode of activation, regulation and inhibition, substrate specificity, and molecular cell function, which underlie differences in the expression pattern of p38MAPK isoforms in different tissue and organs of the body implicated in different infectious disease occurrence [90]. It was reported that p38αMAPK specifically induces the synthesis of proteases, which are essential substrates in the inflammatory process. Hence, there is a need to explore the mechanism of the protease’s biosynthesis as it functions in regulating molecular and cellular processes [105]. However, the molecular target of the pyridinyl imidazole family of compounds suppressing inflammatory cytokine biosynthesis was p38αMAPK [106]. Induction of several other inflammatory molecules such as COX2 and inducible nitric oxide synthase [iNOS] was implicated in the p38 MAPK pathway [107,108]. MAPK cell signaling cascades have been reported to transduce endotoxin impulse in the cell’s nucleus, leading to activation of several cell nuclear factors to stimulate genetic and inflammatory changes.

## 5. Component of Nuclear Factor Kappa Beta [NF–κB] Activation in Endometrium Inflammatory Response

NF–κB transcripts were reported in many inflammatory diseases due to their ability to activate through specialized assigned cell receptors reactivity for bacteria and their ligands invasion to modulate the expression of upstream pro-inflammatory cytokines and chemokines [109,110]. NF–kB signaling nuclear activation stimulates inflammatory reactions in endometritis induced by LPS result in phosphorylation and release of specific NF–kB transcriptional factor [75]. NF–kB regulates genes involved in various immune response processes, including mobilization of innate immune cells, inflammation, maturation of dendritic cells, and lymphocytes’ stimulation. In this way, the NF–kB feature can be combined with other signaling pathways and signaling regulated by the prototypical IkB member proteasomal degradation [IkBα] stimulate the production and release of inflammatory cytokines, resulting in prolonged inflammatory responses. NF–kB canonical and atypical [non-canonical] pathways regulate proteolysis of IkB [an NF–kB inhibitor], and IkB related proteins by the mechanisms which control the transcription of inflammatory genes at the same time [111,112,113]. Signal transductions are triggered by the attachment of bacteria ligands to the cell leading to the activation of the two catalytic subunits [IKK1 and IKK2] and the NEMO regulatory subunit IKK complex [75,114]. Activated IKK phosphorylates IkBα, initiating its lysine-48-linked polyubiquitination and proteasomal degradation primarily via the action of IKK2, enabling associated NF–kB subunits to translocate into the nucleus. The NF–kB cloning consists of NF-kB1 [p50 and its ancestor p105], NF-kB2 [p52 and its ancestor p100], RelA [p65], RelC, and RelB, all of which are distinguished by a possibility of an N-terminal Rel homology stratum [RHD] mandatory for homo and hetero-dimerization in sequence-specific DNA boundaries. Non-canonical NF–kB channels, including p105 and p100, also have the crucial immuno-inflammatory potential of the transcriptional signaling pathways [75,112,115]. A subset of TNF family members and triggers of NF–kB activating kinase activated the p100-mediated pathway [NIK] and IKK1 in sequence. IKK1 [IKKα] phosphorylates p100, initiating its poly-ubiquitination and proteolysis through the proteasome to create p52, which translocated distinctly into the nucleus RelB, activating the alternate NF–kB signaling pathway [111,116]. The triggered NF–κBsubunit p65 dissociated from its intracellular protein IκB-α and translocated from the cytoplasm to the nucleus, where specific target genes are transcribed [117].

IKKα and IKKβ are intracellular protein kinases with high sequence relation, often influenced IKB protein phosphorylation, and serves as convergence points for some transduction pathways leading to NF–KB activation. IKKβ is necessary for the rapid activation of NF–kB by pro-inflammatory signaling cascades. Stimulation of cells contributes to phosphorylation of IKKβ, provides a signal identification for reactive enzymes that marks a rapid proteasomal degradation of IkBs. For an effective immune response, prompt activation of NF–kB is needed, but this response cannot last indefinitely and must be correctly terminated to prevent tissue damage. IKKα/NEMO, a 48 kDa regulatory subunit at its N-terminus with a kinase-binding domain and ubiquitous domain at its C-end [37,118]. IKKγ/NEMO serves as a scaffolding or adapter feature in several signal pathways, and the pathophysiological effects of its defects were seen in reproductive failure in mice [116]. IkBs degradation leading NF–kB dimers activated inside the nucleus to stimulate transcription of the target gene [117].

The IKKβ-dependent pathway is essential for natural immunity activation. The IKKα dependent pathway played an important role in adaptive immunity regulation and lymphoid organogenesis of domestic animals. IKKα attenuated signals by the IKKβ-dependent pathway. IKKβ is most critical for the rapid degradation of NF–kB bound IkBs; IKKα enhanced p100 synthesis, contributing to p52 activation: RelB dimers [117,119]. IKKβ can also engage in a negative feedback loop, downregulating the signaling pathways contributing to its activation. The IKKβ-dependent pathway, known as the classical NF–kB pathway to process p100 and the activation of p52, while RelB is known as an alternative NF–kB pathway [119]. IKKγ is essential for binding, catalytic subunits to upstream activation in the classical NF–κBpathway. IKKα phosphorylation is not essential for most pro-inflammatory response triggers to activate the classical IKK complex or NF–kB activation [120,121]. The molecular activation of NF–κBand its component biomolecules stimulate increased pro-inflammatory cytokines and other genetic factors in the cell nucleus.

## 6. Inflammatory Cytokines and Chemokines in Bovine Endometritis

Shreds of evidence exist that cytokines (including chemokines) are essential in a pregnancy-dependent manner by the bovine endometrium [43,66,122,123,124]. The cytokines and chemokines are key mediators of innate immunity, necessary for the clearance of this infection and resolution of inflammation. They are also necessary for the postpartum endometrial involution and antimicrobial peptides [AMPs] [44,45,46,47]. Cytokine reportedly functioned in cell apoptosis in which they exhibit enormous stimulatory, synergetic, and redundancy interaction among cells. Chemokines were thought to be surfactant cytokines with various biological events in controlling leukocyte trafficking, immune response, cell migration, and growth factors [16,110,123]. The uterus is an immunocompetent site usually poised at defeating microbial infections as soon as they are established during postpartum. Still, its clearance capacity may wear off, causing prolonged or excessive inflammatory response [125,126]. Lee, 2020 [127] reported that AP-1 and IRF-3 in the cell nucleus are considered pivotal factors in regulating inflammation by producing pro-inflammatory mediators and cytokines. The bovine endometrium was reported to express a wide range of immune factors, including interleukin [IL] 1β-2, -6, -8, and -10, and tumor necrosis factors. They also involved cytokines in triggering an immune reaction, acute inflammatory events, and chronic inflammatory changes or changes in uterine flush from cows with endometritis [43,126]. Several scientists have asserted that pro-inflammatory cytokines are necessary for the maturation and disintegration of the follicle, ovulatory process, and corpus luteum development. This evidence might offer an insight into their roles in the pathomechanism of infertility because of subclinical endometritis [40,43,47,126]. Sometimes the pro-inflammatory cytokines are generated and triggered by the association of macrophages with the endotoxin microorganism. Often, they perform important roles in inflammatory disease pathophysiology, including bovine endometritis. The TNF-α is the earliest endogenous mediator of inflammatory responses. IL-1 was the host’s central and secretive mediator of the inflammatory immune response to infections when the endometrial inflammatory response increased [128,129].

IL1 upregulated expression in cows with clinical endometritis seven days postpartum and maintained IL1α and IL1β in cows to 21 days postpartum. Il-1β plays a crucial role in the local and systemic stages of inflammatory reaction. Its early upregulation enhanced and increased endogenous activation of other inflammatory mediators and contributor to cytological endometrial pathophysiology, causing significant tissue damage and septic observations [130]. IL6 is mainly important in inflammatory acute-section reactions. After seven days of parturition, IL6 levels were reliably elevated in apparently healthy cows, cows with clinical endometritis, the cow with dystocia [131], and cervicovaginal mucus [132]. The inflammatory endometrial response elevation of IL-6 is normal to ensure uterine involution [133]. Chemokines improve trophoblast cell migration inhibition and invasion during placental treatment. LPS-stimulated bovine endometritis induced an elevated concentration of chemokines [36].

CXCL8 and CXCL5 function immunologically in uterine tissue infection’s physiological and pathological processes and endometrial cell inflammation [134]. CXCL1 and CXCL6 aid in neutrophil activation, which is associated with mechanisms of inflammation, which apoptosis. In human and canine endometriosis pathogenesis, it was observed that CCL5, CXCL5, and GRO1 chemokines play a role. However, there is a paucity of research information on the pathomechanism of receptors and signal pathways through which activation or upregulation of chemokines occur. Effective immunotherapy needs to be developed to curb incessant cytokine/chemokine-related pathology in subclinical endometritis [135]. The definite connection between cytokine levels and endometrial infectious pathogenesis requires uncompromising research to provide a way out for immunomodulatory treatment and prevention against subclinical endometritis [136].

## 7. Application of Nanotherapeutic in Reproductive Diseases

Due to recent developments in terms of excessive resistance and abuse of antibiotics, the effect in the reproductive tracts of production animals has signaled the discovery of the concept of nanomedicine in livestock development [137]. In the area of infection control of veterinary medicine, nanotechnology has a promising role in preventing or treating infections [137,138,139]. Nanoparticles may present a feasible alternative to antibiotics and help bar pathogens from entering the endometrium at postpartum. More focus has been placed on organic nanoparticles’ use in reproductive medicine due to their biocompatibility and biodegradability [140]. Organic nanoparticles can either encapsulate the drug inside or integrate the drug on the nanoparticle’s surface [141]. A major advantage with nanotherapy is the ability to perform surface modification, conjugating the nanoparticle to targeting peptides or antibodies [142]. There have been many studies about the antimicrobial activity of polymeric materials. Nanoscale systems could contribute greatly to improving innovative therapies for infectious reproduction diseases because of their tunable size and increased suspendibility and surface tailorability, which enhances interactions with biological systems at the molecular level [139,143]. For example, studies have shown that silver ions can make structural changes in the cell membrane [144]. Silver has a high affinity for negatively charged side groups on biological molecules such as sulfhydryl, carboxyl, phosphate, and other charged groups distributed throughout microbial cells. Silver ions inhibit several enzymatic activities by reacting with electron donor groups, especially sulfhydryl groups [145,146]. Silver ions induce the inactivation of critical physiological functions such as cell wall synthesis, membrane transport, nucleic acid (RNA and DNA) synthesis, translation, protein folding and function, and electron transport [144,146]. Kim et al. [147] found that silver nanoparticles could inhibit the growth of hemorrhagic enteritis-inciting *E. coli* O157:H7 and yeast isolated from a case of bovine mastitis. Iron oxide nanoparticles have previously been used as a thermal ablation treatment [148], and it has been suggested that they could form the future basis for hyperthermia-related endometriosis treatment. Cerbu et al. 2021 [149] sought to control the release of tilmicosin by using hydrogenated castor oil–solid lipid nanoparticle carriers in mastitis.

Poly (ϵ-caprolactone) (PCL) lipid-core nanocapsules (LNC) are a promising drug/bioactive compound carrier with a high potential for biomedical applications due to their biodegradability and biocompatibility characteristics. LNC is a vesicular carrier comprising a core structured by a dispersion of solid lipids (sorbitan monostearate) and liquid lipids (caprylic/capric triglyceride), creating a PCL-surrounded polymeric wall [150].

In 2013, Tu’uhevaha et al. [151] reported that EGFR-targeted EnGeneIC Delivery Vehicles (EDVs) loaded with doxorubicin significantly inhibited trophoblastic tumor cell growth in vivo and in vitro and induced significant cell death ex vivo, potentially mediated by increasing apoptosis and decreasing proliferation. EDVs may be a novel nanoparticle treatment for ectopic pregnancy and other disorders of trophoblast growth to militate against Ectopic pregnancy.

Polymeric biodegradable form of Engineered Nanoparticles (pbENPs) has been proposed as effective platforms for the protection and controlled release of reproductive hormones, including steroid or gonadotropic hormones [152]. A previous study of chitosan nanoparticles on hCG (Human Chorionic Gonadotrophin) hormone increased dairy cattle ovulation induction [153]. There was the dissolution of endometrial tumor cell in endometrial cancer in a polymeric nanoparticle (NP) delivery system, which improves efficacy and safety of the combinatorial strategy [154] and also the development of a targeted drug delivery system for the uterus utilizing an immunoliposome platform targeting the oxytocin receptor leading to targeted liposomal drug delivery to the myometrium is reduced dose and reduced toxicity to both mother and fetus [155]. Likewise, Hassanein et al. 2021 [156] reported the fabrication of Gonadotropin-releasing hormone (GnRH)–loaded–Chitosan Nanoparticles could allow a reduction in the conventional intramuscular GnRH dose used for AI in rabbits to half without affecting fertility. The combinatorial effect, poly(lactic-co-glycolic) acid nanoparticles loaded with epigallocatechin gallate(EGCG) and doxycycline (Dox) in a single vehicle appears to be promising for treating endometriosis. The significant decreases in endometrial glands and microvessel density in the Dox-EGCG NP-treated group compared to the groups treated with Dox NPs and EGCG NPS confirmed the increased efficacy of Dox-EGCG NPs compared to the single drug-loaded nanoparticles [157]. Thus far, the literature examining nanoparticles and nanotherapy in subclinical endometritis has been sparse, calling for research concerns.

## 8. Conclusions and Perspectives

The purpose is to treat, control and prevent lingering subclinical endometritis, which has been a significant bottleneck to the development of the dairy industry. The roles of Gram-positive bacteria and their ligands potentials in the pathomechanism of bovine endometritis need further exploration. The cell receptor and their biomolecular component were the primary cell wall receptors point of this review and found crucial research loopholes in the viability of cell receptors that may be distinct for multifaceted research development. Several TLRs have been reported; TLR4 is the most evaluated with little or no record on the molecular activities of other TLRs in the pathogenesis of endometritis. The possibility of heterodimerization within TLRs was reported between TLR2/4/6 exhaust much interaction between the cell receptors, which may be a good clue in drug discovery. TLRs cytosolic adaptor proteins such as MyD88, IRAK1/2/4, TRAF6, TIRAP, TRAM, and SARM, are essential in the inactivation and potentiation of TLRs during stimulation by bacteria PAMPs. NLRs are specialized intracellular or cytoplasmic receptors, enhanced the innate immune function of bovine endometrium in early postpartum, but their function was compromised due to massive invasion of microorganism. The NLRP inflammasomes also serve as the hallmark of inflammation in immune cells, and there are several kinds of it; only NLRP3 inflammasome was bit explored in the molecular pathogenesis of bovine endometritis.

Other NLRs need to be investigated [52,84]. The mode of activation, regulation, and inhibition, substrate specificity, and molecular cell function underlies the difference in the expression pattern of ERK, p38MAPK, and C-JNK isoforms in bovine endometrium need further molecular and immunological characterization [158]. The MAPKinhibitors are promising therapeutic agents in inflammatory diseases in which bacterial products and pro-inflammatory cytokines play a critical role in their pathogenesis. The neglect of some NF–kB protein kinases and their signaling subunits in evaluating and regulating serial cascades in the molecular pathogenesis of dairy cow postpartum endometritis call for concern [159]. The therapeutic invention of NF–kB and IKK inhibitors for treating inflammation and cancers, some of which are under clinical trials (Figure 3). The cellular physiological role of IKK-induced p105 needs to be established in the proteolysis of NF–kB activation in the pathogenesis of bovine endometritis [117].

Numerous studies have demonstrated that cell biomolecules are good delivery platforms that increase the uptake of antigens and adjuvant, leading to better immune responses [160,161,162]. Understanding the molecular and cellular mechanism underlying the pathogenesis of postpartum endometritis may help in immunotherapies or nanotherapeutic discoveries for preventing and regulatingthe menace of dairy cow uterine disease. With the strong demand to develop alternative therapeutic options to address unrealized therapeutic needs, novel nanotechnology-based platforms have recently provided an important baseline for cell health and protection.The development of nano-based systems has provided protection strategies for incorporated agents, such as biomolecules—nucleic acids, peptides, and proteins which are generally quickly degraded when administered in vivo. 

The development of nano-based systems based on cell particles has also been described as platforms for targeting and delivering therapeutic agents and nanodevices and analytical systems for theragnostics. The range of applications of nanosystems can include drug delivery, cancer, gene therapy, and imaging and cell tracking through biomarkers and biosensors [163], thereby allowing for the use of prophylactic measures to avoid the progress of the disease or to greater efficacy of therapies due to an earlier treatment [162]. Therapeutic agents can be embedded, encapsulated, or even adsorbed or conjugated onto the nanosystems, which can be modified and associated with other cell biomolecules to achieve an optimized release profile [164,165]. Future in-depth research into the series of cell signaling, sensors, effectors, and biomolecules of the bovine endometrium that activate the pathomechanisms of infection is essential in developing an alternative nanotechnology-based therapy against endometritis in dairy cows, as it was recently found to cure some diseases. During the writing of the manuscript, more than 308 published articles were reviewed. In the 144 articles relevant to the molecular exploration of the pathogenesis of bovine endometritis, nanotherapeutic discoveries were evaluated. The articles published in the last and these decades were selected for the write-up. The journals selected were science citation index journals with good impact factors. Most articles have wide views by different researchers, which means most the article is within the researchers’ reach.

## Figures and Tables

**Figure 1 animals-11-01516-f001:**
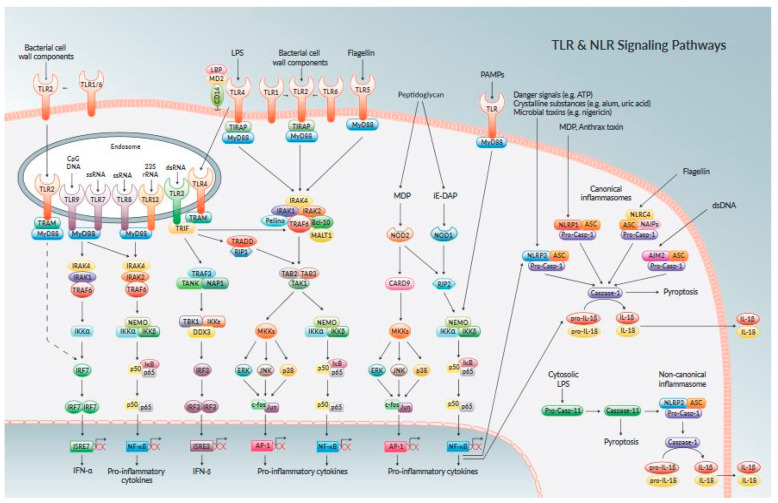
The mechanism of pathogen molecular recognition by Toll-like receptors triggering cellular signaling pathways activation (26,48). The mechanism of PAMP stimulating the cell receptor with their major transduction by biomolecules, which induced various isoforms of the signaling pathway and led to upregulation of pro-inflammatory cytokines and chemokines. The delivery of nanodrugs in suppressing inflammatory response in any part of the signaling pathways is essential in nanotherapeutic development against endometritis.

**Figure 2 animals-11-01516-f002:**
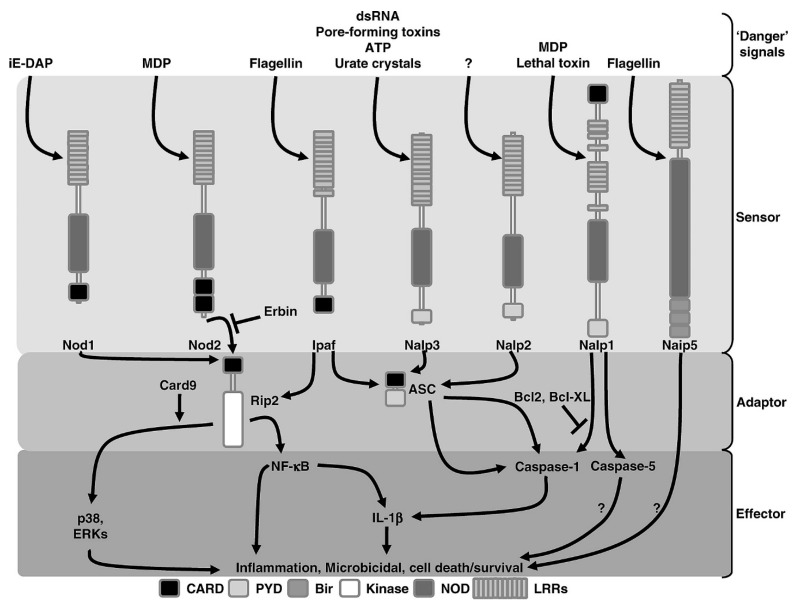
Intracellular Debugging of the NLR Signaling Pathways showing the cellular interconnectivity between PAMPs, cellular sensors, adaptors, and effectors to activate intracellular signaling cascade and stimulation of inflammation ultimate. The intracellular connections between its biomolecules are the basis for nanomedical and nanotechnology discovery against lingering subclinical endometritis.

**Figure 3 animals-11-01516-f003:**
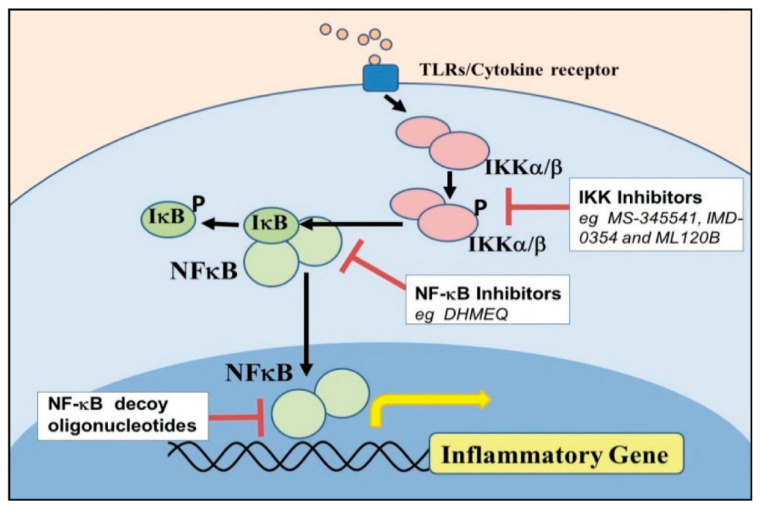
Schematic diagram narrating the potential cascade of NF–κBand IKKαβ inhibitors to protect inflammatory gene.Table 88. Myeloid Differentiation factor 88.

## Data Availability

Not applicable.

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
