# Peer review of "Modulation of Bovine Endometrial Cell Receptors and Signaling Pathways as a Nanotherapeutic Exploration against Dairy Cow Postpartum Endometritis"

_animals, 2021, doi:10.3390/ani11061516_

Round 1

Reviewer 1 Report

This manuscript reviews several molecular mechanisms involved in the subclinical endometritis in cattle. This is a narrative review therefore it has an intrinsic risk of bias regarding the literature included, however it is a good exercise and it is of interest for the practitioners and academics involved in the area. However, one suggestion would be to include (as supplemental information), the strategy followed to obtain the information included and if there was some quality control carried out to ensure the relevance and validity of the data presented.  This would greatly increase the relevance of the paper and would strengthen the information present.

The information included is relevant and thorough but because of the nature of the paper delving into the molecular aspects, a clear description and cohesive order is required, so a suggestion to the authors is to check the English used in the manuscript to ensure that it is easy to read and clear to the readers.

Overall, I support the publication of the manuscript following a major revision of the style and inclusion of the strategy followed to select the information presented.

Author Response

Dear Reviewer, 

Glad to hear from you, the list comment has greatly help us to improve the manuscript. Supplementary information has been insert in the manuscript. English language and cohesiveness has improved.

Reviewer 2 Report

The manuscript presented by the authors reviewed the cell receptor and relevant signaling transduction pathways involved in the pathogenesisi of bovine endometritis, and posed an nanotechnology-based drug delivery system against persistent endometritis. In general, it is of importance to  develop new approaches to treatment, prevention, and control of bovine endometritis. However, there are some concerns need to be addressed.

1. English language and style require extensive editing.

2. Line 64-66: “Several diagnostic, treatment and prevention have been adopted to eliminate this postpartum uterine infection, which includes rectal palpation of the genital tract to detect uterine abnormalities”, Please detail the latest diagnostic, treatment, and prevention measures.

3. The abbreviation of Escherichia coli  should be corrected in the text (Line 138, Line 142).

4. Line 160: Mycoplasma species belong to the Gram-positive bacteria. Please check and revise it.

5. Line 235-236: “TLR4 is the most examined receptors in bovine endometrium, and their messenger. RNA expression”. The “.” should be deleted.

6. Line 244: “common domain organization with aNH2- terminal protein-protein interaction domain”. Please write it correctly.

7. Line 332-342, The authors mentioned NLRP5, NLRP6, but did not describe whether these inflammasome are related to bovine endometritis.  Please explain it.

8. Line 466: “The IKKα dependent pathway is more important for adaptive”. Please write it correctly.

Author Response

Dear Reviewer, 

The English language and style of the manuscript has been greatly improved. All the concerns listed before has been addressed in the revised version of the article.

Reviewer 3 Report

Dear Dr. Zhang

Assistant Editor of Animals

The manuscript describes the knowledge and possible applications on the Modulation of bovine endometrial cell receptors and signaling pathways as a nanotherapeutic exploration against dairy cow postpartum endometritis.

The review is well structured in section, all aimed at explaining the infimation process at the endometrial level. In fact, it starts from the action of bacteria on endometrial cells up to the inflammatory response of these cells. Furthermore, the cellular receptors and all cellular mediators of the inflammatory message are described with great precision.

Finally, an innovative approach is proposed for the treatment or prevention of endometritis which are the cause of dysfunctions affecting the reproductive system of cows.
Ultimately, the review takes stock of the knowledge on this topic and gives ideas for developing new research on how these concepts can be applied.

Therefore, the manuscript for its good quality, in my humble opinion, deserves to be published in this Journal.

Kind regards

Author Response

Dear Reviewer,

Great and cool to receive your comment on our manuscript. We have improved on the structure and cohesiveness of the article and proposed innovative strategies for the treatment and diagnosis of bovine endometritis

This manuscript is a resubmission of an earlier submission. The following is a list of the peer review reports and author responses from that submission.